# Estimating the replicability of highly cited clinical research (2004–2018)

**Gabriel Gonçalves da Costa**[ORCID]*, **Kleber Neves, Olavo Amaral**

Institute of Medical Biochemistry Leopoldo de Meis, Federal University of Rio de Janeiro, Rio de Janeiro, Rio de Janeiro, Brazil

* gabriel.costa@bioqmed.ufrj.br

## Abstract

### Introduction

Previous studies about the replicability of clinical research based on the published literature have suggested that highly cited articles are often contradicted or found to have inflated effects. Nevertheless, there are no recent updates of such efforts, and this situation may have changed over time.

### Methods

We searched the Web of Science database for articles studying medical interventions with more than 2000 citations, published between 2004 and 2018 in high-impact medical journals. We then searched for replications of these studies in PubMed using the PICO (Population, Intervention, Comparator and Outcome) framework. Replication success was evaluated by the presence of a statistically significant effect in the same direction and by overlap of the replication's effect size confidence interval (CIs) with that of the original study. Evidence of effect size inflation and potential predictors of replicability were also analyzed.

### Results

A total of 89 eligible studies, of which 24 had valid replications (17 meta-analyses and 7 primary studies) were found. Of these, 21 (88%) had effect sizes with overlapping CIs. Of 15 highly cited studies with a statistically significant difference in the primary outcome, 13 (87%) had a significant effect in the replication as well. When both criteria were considered together, the replicability rate in our sample was of 20 out of 24 (83%). There was no evidence of systematic inflation in these highly cited studies, with a mean effect size ratio of 1.03 [95% CI (0.88, 1.21)] between initial and subsequent effects. Due to the small number of contradicted results, our analysis had low statistical power to detect predictors of replicability.

### Conclusion

Although most studies did not have eligible replications, the replicability rate of highly cited clinical studies in our sample was higher than in previous estimates, with little evidence of

**Data Availability Statement:** All pertinent data related to this study are openly accessible through the Open Science Framework (OSF) Repository at the provided URL: https://osf.io/a8zug/. Further

comprehensive information regarding data, code, and analyses associated with this research can be located within both the preprint (https://doi.org/10.1101/2022.05.31.22275810) and the manuscript presently under evaluation for publication in your esteemed journal. It is noteworthy that all the aforementioned elements are conveniently housed within an Open Science Framework repository. Furthermore, it is of significance to underscore that this study was subjected to preregistration on the OSF platform, and relevant details pertaining to this aspect can be accessed through the previously mentioned links. Additionally, Supporting information has been appended to this submission, and all the data, tables, graphics, and figures corresponding to this supplementary material are publicly available on the OSF platform as well.

**Funding:** The author(s) received no specific funding for this work.

**Competing interests:** The authors have declared that no competing interests exist.

systematic effect size inflation. This estimate is based on a very select sample of studies and may not be generalizable to clinical research in general.

## Introduction

The replicability of published research has been recently questioned in different scientific fields, with replication rates shown to be variable and often low [1–8]. Whether this represents a "reproducibility crisis" is open to debate [9], and defining what constitutes a successful replication is not trivial [10]. Systematic replication efforts have mostly focused on restricted samples of the literature, and data on the subject is still lacking in many areas.

The replicability of highly cited clinical research was studied by Ioannidis in 2005, based on available published replications of a sample of articles between 1990 and 2003 [11]. It focused on the reproducibility of study conclusions, typically assessed by statistical significance, as well as on effect size comparisons. 44% of highly cited studies had been successfully replicated, 16% had been contradicted, 16% had found effects that were larger than those of subsequent studies, and 24% remained unchallenged.

A similar effort for highly cited psychiatry research between 2000 and 2002 found lower estimates, with 19% of studies replicated, 19% contradicted, 13% initially inflated and 48% unchallenged [12]. Another study on critical care research found that 18% of interventions published in high-profile journals between 1946 and 2016 had their results replicated by a subsequent study, whereas 22% were contradicted, 2% had replications in progress and 58% remained unchallenged [13].

Clinical research has changed in some aspects over the last two decades. A priori registration of study protocols has become more common and mandatory for clinical trials in many countries [14]. Although publication bias has not been eliminated [15], the likelihood of null results has increased in published studies [16]. Reporting guidelines have become more widely used and underwent reevaluations and updates [17, 18]. The push for full reporting of results and availability of individual patient data has also gained ground [14, 19]. Thus, the replicability panorama in high-impact clinical research may have changed during this period [20–22].

In light of this, the goal of this study is to estimate the replicability of highly cited clinical studies published between 2004 and 2018. Our primary outcome is the rate of successful replication in these studies, as measured both by statistical significance in the same direction and by overlap of CIs for the main effect measure in both studies. We also explore effect size inflation and potential predictors of replicability.

## Methods

An overview of the project, datasets and analysis code can be found at https://osf.io/a8zug/. The protocol for the study was preregistered at https://osf.io/nh965/, with a step-by-step methodology available at https://osf.io/2qncz/ and updates and amendments described at https://osf.io/26d98/. All statistical analyses were performed in R version 4.1.2 [23]. Data and analysis code are available at https://osf.io/5qhdz and https://osf.io/9hmx4, respectively.

### Search for highly cited studies

We searched the Web of Science database for articles with more than 2000 citations published between January 1st, 2004, and December 31st, 2018, in medical journals with an impact factor above 14 in the 2020 Journal Citation Reports (list at https://osf.io/2qncz/). General journals

were searched on February 7th, 2020, and specialty journals on March 4th, 2020. The cutoffs for citation and impact factors were twice as large as those used by Ioannidis [11], accounting for the growth in the total number of articles in PubMed during the period (calculation at https://osf.io/t9xu7/).

Within this sample, one author (K.N.) screened titles and abstracts for articles that addressed the efficacy of therapeutic or preventive interventions with primary data (i.e., excluding reviews, meta-analyses or articles that combined two or more previous studies). Two evaluators (G.G.C. and K.N.) then selected the primary outcome in each study, or the main conclusion in the abstract if the study had no primary outcome. In the case of co-primary outcomes or equally emphasized conclusions [11, 24–26], we chose the outcome that was deemed more clinically relevant (e.g., mortality over progression or neurologic improvement over reperfusion). In the case of trials with more than two arms [27], we selected the most effective drug as the intervention and randomly chose an active comparator. Studies with no control group (e.g., phase 1 trials) were considered eligible if the abstract clearly stated that an intervention was clinically effective. When both benefits and harms or caveats were presented, focus was given on the net conclusion of whether the experimental intervention merited consideration for use in clinical practice. Disagreements in outcome selection were solved by consensus with the help of a third investigator (O.B.A.).

For each article, the study design, sample size, journal name and category (general or specialty) were extracted. We also extracted the selected outcome measure–i.e., odds ratio (OR), relative risk (RR), hazard ratio (HR), incidence rate ratio (IRR) or objective response rate (ORR)–with its effect size and respective CI. For controlled studies, results were classified as positive or negative according to the authors' stated statistical significance threshold. Non-inferiority trials were classified as positive only if the intervention was found to be superior (i.e., not merely non-inferior) to the comparator.

For each result, the population, intervention, comparator and outcome (PICO) [28], both in specific (e.g., "myocardial infarction, ischemic stroke, unstable angina, or cardiovascular surgery") and general forms (e.g., "cardiovascular events") [14] were extracted. Two evaluators (K.N. and G.G.C.) described PICO components independently and resolved disagreements by consensus. Results of the independent extraction and consensus decisions can be found at https://osf.io/sfdxv.

## Search for replications

After agreement was reached on PICO components, two evaluators (G.G.C. and K.N.) performed independent searches for replications of highly cited studies in PubMed. Search terms were defined independently by each evaluator and included the name of the drug or intervention, the general form of the outcome, and the population (i.e., clinical condition) as described in the article's title, along with corresponding Medical Subject Headings (MeSH) terms. The comparator was included in the search strategy only if it was an active intervention (i.e., not a placebo or sham). Details can be found at https://osf.io/zv65u/.

A study was considered a replication of the highly cited study when it shared the same PICO general components, namely (a) the drug or intervention, without considering dose or regimen (except for studies performing dose or regimen comparisons), (b) the general form of the outcome (as described in [14]), (c) the population/clinical condition as described in the highly cited article's title and (d) the comparator. When geographical information was included as a descriptor of the population (e.g., "European patients") [29, 30], we did not include this information as part of the population component [31–33].

Replications needed to be (a) a study type with higher strength of evidence [34] (i.e., randomized controlled trials (RCTs) over cohort studies over smaller uncontrolled studies) or (b) a similar study type with a sample size equal to or larger than the original study. Meta-analyses were considered as eligible replications if the highly cited study accounted for less than half of their sample size. For network meta-analyses, only the sample size of the direct comparison between the intervention and comparator counted for this purpose. If a meta-analysis [35, 36] included a single additional study beyond the highly cited one [32, 37], we considered the effect size of this study as the replication [31, 38], rather than that of the entire meta-analysis. If more than one replication was found, the one with the largest sample size for the specific comparison was considered.

When different replications were selected by each evaluator, both were made available for the two evaluators to choose the best option independently. Disagreements in this step were solved by consensus with the participation of a third author (O.B.A). Agreement in the initial selection was 36%, but rose to 91% when selected replications were made available to both evaluators. Agreement data can be found at https://osf.io/qz6u9, with resolution of disagreements detailed at https://osf.io/ma9bn.

After identifying the best available replication, both evaluators independently selected the outcome and effect size from the replication that corresponded most closely to the one in the original study. Disagreements were solved by consensus. For network meta-analyses, direct comparisons were favored over indirect ones when both were available, either in the manuscript or supplementary material. Agreement data for this process can be found at https://osf.io/2c6jx. Changes in the choice of replication and effect size during analysis are documented at https://osf.io/jq7ec.

As the effect estimates of meta-analyses usually included the highly cited study and were thus not fully independent from it, we re-estimated these effects after removing the highly cited when enough information was provided for this purpose. For this, we used the primary study results as retrieved from the meta-analysis, estimating effect sizes based on numbers of events and patients when these were available, or the log-transformed point estimate of the RR or HR when they were not, with a standard error estimated by $\frac{[\log(\text{upper limit of CI}) - \log(\text{lower limit of CI})]}{3.92}$. For data synthesis, a random-effects model was performed using the Mantel-Haenszel method for effect size estimation in the package meta in the R software for statistical computing [39]. Replicability rates using the effect sizes from these fully independent meta-analyses are provided in addition to the main results as a supplementary analysis.

## Evaluating replication success

Pairs of highly cited studies and their replications were analyzed to evaluate whether results were successfully replicated on the basis of two criteria: (a) statistical significance (an effect in the replication with $p < 0.05$ in the same direction as that observed in the highly cited study) and (b) confidence interval overlap (an overlap of the 95% CIs for the outcome of interest in both studies). When the highly cited study presented a non-significant effect or did not include a statistical comparison (e.g., phase 1 trials), only the second criterion was used. The primary outcome was the rate of successful replication in our sample by both criteria (or by CI overlap alone when statistical significance was not applicable). As additional criteria, we analyzed whether the replication point estimate was contained in the 95% CI of the highly cited study and vice versa. A sensitivity analysis was performed applying the statistical significance criterion to initially non-significant studies as well. In one case where the replication was a Bayesian meta-analysis, the original study's CI was compared to a credible interval (CrI), in the case of minimally informative priors [40]. P-values were calculated from effect sizes [point estimates

and confidence intervals] for each replication and highly cited study (details at https://osf.io/jbn83).

When outcome measures differed between highly cited studies and replications (e.g., RR in the highly cited study vs. OR in the replication or vice-versa) and the replication was a primary study, the replication measure was converted to the one in the highly cited study using the data available in the article. When the replication was a meta-analysis that included the highly cited study, we chose the risk measure that was used for data synthesis, using the original study's effect size as included in the meta-analysis (details at https://osf.io/rfqgd). If the highly cited study was not included in the meta-analysis (e.g., when the meta-analysis included an update of the highly-cited study with a longer follow-up), we manually converted the outcome measure of the highly cited study to the one in the meta-analysis using the original data. When the original study was a phase 1 trial measuring ORR, we manually calculated this measure in replications when needed, with CIs based on the Clopper-Pearson exact method. In the meta-analysis by Hamid et al. [41], ORRs for both RCTs that were eligible replications of the highly cited study (a phase 1 trial) were calculated manually based on the combined data. Details can be found at https://osf.io/mfwv2.

95% CIs for replicability rates were calculated by $p \pm 1.96 \sqrt{\frac{p(1-p)}{n}}$. Replicability rates in the main results use effect sizes from published replications, while supplementary analyses use only fully independent replications, recalculating meta-analytic effect sizes in the absence of the highly cited study (and excluding meta-analyses for which this was not possible).

## Effect size inflation

Effect size inflation was estimated on the basis of ratios between the effect sizes of highly cited studies and replications. For unfavorable outcomes (e.g., death, tumor progression), in which effectiveness increases as the outcome measure decreases, the inflation ratio was defined as the point estimate of the replication divided by that of the original study. For favorable outcomes (e.g., neurologic improvement), in which effectiveness increases along with the outcome measure, it was defined as the point estimate of the original study divided by that of the replication. Publication order was considered in this calculation: thus, when the replication was a meta-analysis in which the pooled sample size of studies preceding the highly cited study was larger than that of those that followed it, we inverted the ratios, considering the highly cited study as the replication and vice-versa for this purpose. This was also performed in a case where the replication was an RCT identified within a meta-analysis [37] that was published before the highly cited one [31]. CIs for mean effect size inflation were calculated by the Wilson score interval.

As these two adjustments had not been pre-specified in the protocol, we performed sensitivity analyses using different ways to deal with positive/negative outcomes and study order within meta-analyses when analyzing effect size inflation. For the former, coining of the effects was performed to convert all favorable outcomes to unfavorable outcomes: objective response rates were subtracted from 1 (1 –ORR), odds ratios were inverted (1/OR) and relative risks for the complementary outcome were calculated based on the original data (https://osf.io/5tmus and https://osf.io/7wtr8). For the latter, we analyzed data considering the highly cited study as the original one, independent of study order in the meta-analysis. As done for replication rates, we also provide supplementary effect size inflation analyses based on fully independent replications only, using recalculated meta-analytical estimates in the absence of the highly cited study.

For analysis of effect size inflation, natural logarithms of the ratios were used for each study pair (including those with initially negative results) to calculate the mean and CI of these

ratios, both for the whole sample and for phase 1 trials and RCTs separately. For the whole sample, we performed a one-sample t-test against a theoretical mean of 0, which would indicate absence of systematic inflation. Although these calculations were performed using log-transformed values to correct for the inherent asymmetry of ratios, we transformed means and CIs back to a linear scale for clarity when describing results.

### Predictors of replicability

Finally, we analyzed if studies with contradicted results–i.e., those failing in one or both replication criteria–differed from successfully replicated ones in the following aspects: (a) study design (RCTs vs. other designs); (b) nature of intervention, (pharmacological vs. non-pharmacological); (c) sample size; (d) p-value of the original study; and (e) citations per year. To compare these aspects between replicated and contradicted studies, we used Fisher's exact test for categorical variables (a and b) and Mann-Whitney's U test for continuous variables (c through e). We had also planned to use the effect size of the highly cited study as a predictor, but due to the heterogeneity in outcome measures, which included both proportions (i.e., objective response rates) and measures of association (i.e., ORs, RRs, IRRs and HRs) converting them to a single effect size measure turned out to be unfeasible.

## Results

Results from our systematic search of the literature are shown as a flowchart in Fig 1. A total of 89 highly cited studies met our inclusion criteria. Of these, 24 had an eligible replication according to our criteria.

As shown in Table 1, included studies received a median of 2842 citations, and were mostly RCTs of pharmacological interventions in cancer or heart disease, with some phase 1 cancer trials as well.

Most replications were direct-comparison meta-analyses, followed by RCTs, network meta-analyses and a phase 2 trial (Table 2), with RCTs more commonly representing replications of phase 1 trials. Two meta-analyses [42, 43] replicated more than one highly-cited study in the sample (2 each). All phase 1 trials had available replications in the literature, while the only cohort study in our sample had no eligible replication.

A list of claims from highly cited studies is shown in Table 3. With few exceptions, most of them made claims of efficacy in their abstracts. Efficacy in phase 1 trials was measured by ORR (n = 7), while differences in outcomes in RCTs were measured by HR (n = 8), RR (n = 5), IRR (n = 1) or OR (n = 3). All phase 1 trials made clear claims of efficacy based on tumor regression. Among RCTs, 2 were negative, with a p-value above the standard cutoff of 0.05.

When a meta-analysis was selected as a replication, the highly cited study could come after some or most of the studies in the meta-analysis, and thus consist of a replication of previous literature itself. Fig 2, shows the relative sample sizes of the highly cited studies in their replications. On average, highly cited studies corresponded to 18% of the total sample size of the meta-analyses, but this number ranged from 2% to 42% (samples sizes can be found at https://osf.io/ma9bn). Most meta-analyses had a larger number of patients after the highly cited study than before it, with three exceptions [30, 33, 66], including one [42] in which all other studies in the meta-analysis preceded the highly cited one [33]. In another case, a meta-analysis of 2 studies [37] led to an RCT published before the highly cited study [31] to be selected as a replication.

Replication rates using different criteria are shown in Table 4. Among the 15 highly cited studies with statistically significant results, only 2 (13%) had a non-significant result in the replication, whereas the 2 highly cited studies with negative results had significant results in their

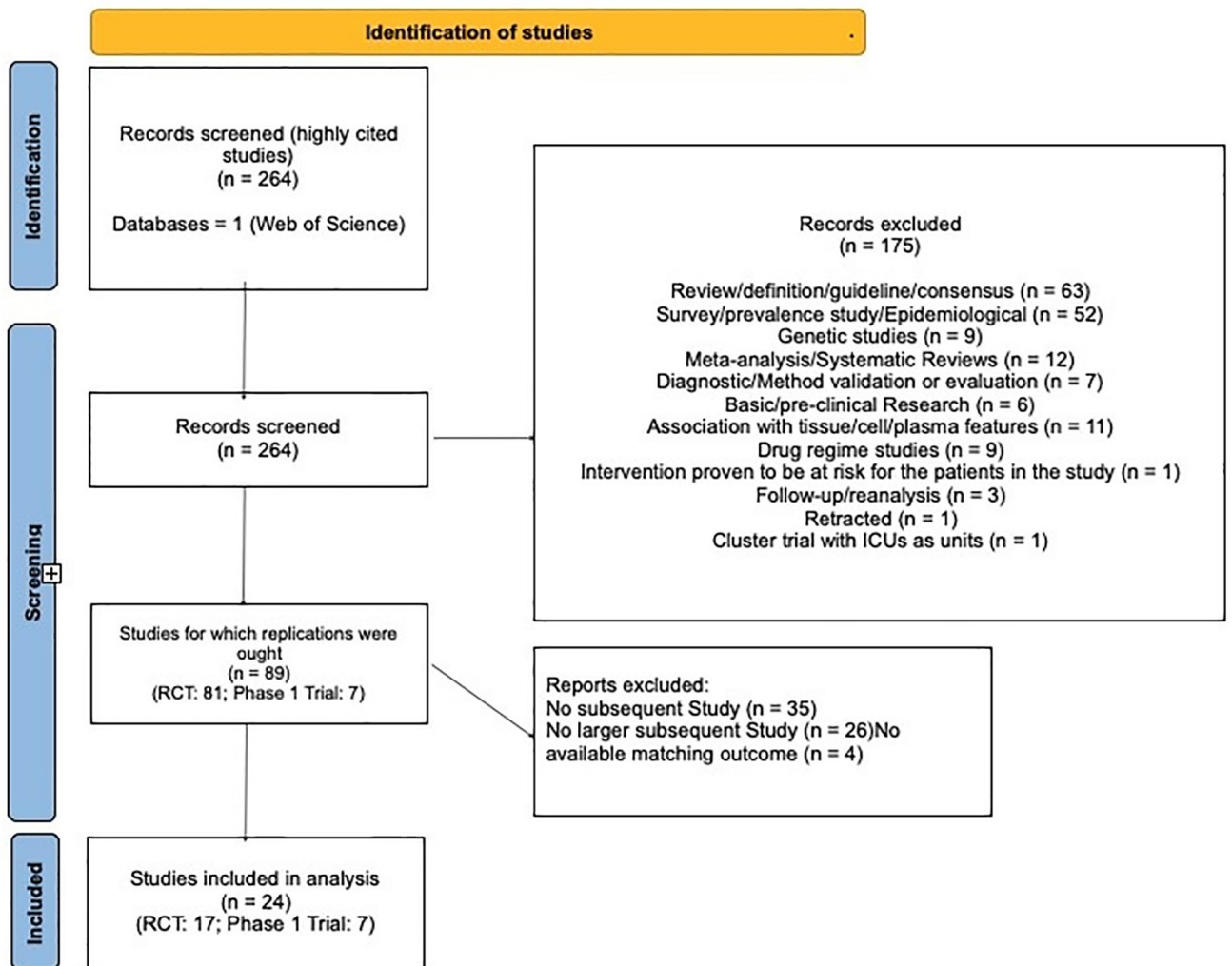

**Fig 1. PRISMA flowchart.** 89 eligible highly cited studies were found, of which 24 had an eligible replication. A complete list of the studies can be found at https://osf.io/ub38r/. A more detailed list of reasons for inclusion/exclusion is available at https://osf.io/ma9bn/.

replications (albeit marginally so). We did not consider the latter as replication failures in our main analysis, as lack of significance in null hypothesis tests should not be taken as evidence of equivalence, especially considering that sample size was higher in the replications. However, we did perform a sensitivity analysis using the statistical significance criterion for studies with non-significant results as well.

Phase 1 trials with no control group were also not considered for the statistical significance criterion. That said, among the 7 phase 1 trials, 6 had replications with statistically significant results when comparing the intervention to a control group in a RCT; the remaining one was replicated by an uncontrolled phase 2 trial [75].

Concerning the effect size criterion, 21 out of 24 studies (88%) had overlapping 95% CIs, with 2 phase 1 trials and 1 RCT failing this criterion. In total, 15 out of 24 replications (62%)

**Table 1. Features of highly cited studies.** Columns show the numbers of highly cited studies and of those for which an eligible replication was found in each category. Percentages refer to the total number of highly cited studies (n = 89) or studies with replications (n = 24), respectively.

| Description | | Total (n = 89) | % | Replication found (n = 24) | % |
|---|---|---|---|---|---|
| Highly cited study design | RCT | 81 | 91% | 17 | 71% |
| | Phase 1 trial | 7 | 8% | 7 | 29% |
| | Cohort study | 1 | 1% | 0 | 0% |
| Type of intervention | Pharmacological | 72 | 81% | 16 | 67% |
| | Non-pharmacological | 17 | 19% | 8 | 33% |
| Journal | *New England Journal of Medicine* | 80 | 90% | 22 | 92% |
| | *Lancet* | 4 | 4% | 0 | 0% |
| | *Lancet Oncology* | 4 | 4% | 2 | 8% |
| | *JAMA* | 1 | 1% | 0 | 0% |
| Condition | Cancer | 53 | 60% | 15 | 63% |
| | Cardiovascular | 27 | 30% | 8 | 33% |
| | Other | 9 | 10% | 1 | 4% |
| Citations | < 3,000 | 45 | 51% | 13 | 54% |
| | 3,000–4,000 | 26 | 29% | 8 | 33% |
| | 4,000–6,000 | 12 | 13% | 1 | 4% |
| | > 6,000 | 6 | 7% | 2 | 8% |

had point estimates that were contained in the CIs of the highly cited studies; conversely, only 9 (38%) of the original point estimates were included in the replication's CIs. That said, as CIs get narrower with increasing sample size, the latter criterion is excessively strict and should not be considered as a measure of replicability.

One major limitation of this analysis is that effect sizes from meta-analyses are not fully independent from the highly cited study when it is included in the estimate. To circumvent this, we recalculated meta-analytical effect estimates in the absence of the highly cited studies (S1 Table). This was technically feasible without re-extracting data from the original studies for 10 out of 14 meta-analyses (replicating a total of 11 highly cited studies). The remainder were network meta-analyses without direct comparisons [42, 50, 59, 67] or individual patient-data meta-analyses [62].

When using only fully independent replications (i.e., excluding meta-analyses that could not be reanalyzed), replicability rates were 80% for the statistical significance criterion, 84% for the CI overlap and 79% for the aggregated criterion (S2 Table). Differences between this and the main analysis were due to the different samples used in each of them, as no meta-analysis changed its replication status in either criterion when reanalyzed without the highly-cited study.

**Table 2. Features of replications.** Percentages refer to the total number of highly cited studies in each category. For studies with a replication, the table describes the type of study; most studies, however, did not have a valid replication. A network meta-analysis including a direct comparison for the groups of interest [44] was classified as a regular meta-analysis. Two meta-analyses [42, 43] replicating two highly cited studies are counted twice in their respective categories.

| | | Replication study design | | | |
|---|---|---|---|---|---|
| Design of highly cited study | Replication found (%) | Phase 2 Trial | RCT | Meta-analysis | Network meta-analysis |
| Cohort study | 0/1 (0%) | 0 | 0 | 0 | 0 |
| Phase 1 trial | 7/7 (100%) | 1 | 3 | 2 | 1 |
| RCT | 17/81 (21%) | 0 | 3 | 10 | 4 |
| Total | 24/89 (27%) | 1 | 6 | 12 | 5 |

**Table 3. Summary of the 24 highly cited studies with replications.** Table shows the references for both studies, the general PICO components, the original study's conclusion, the outcome measure and the effect sizes (with CIs) found in the original study and replication. Replication studies for PARTNER A [35] and NEJSG [32] were obtained from 2-study meta-analyses found in our search [36, 37] and their effect sizes are drawn from these meta-analyses; in the case of IPASS [31], this corresponds to a subgroup matching that of the highly cited study. Two pairs of studies [MR CLEAN [63] and ESCAPE, Cheng et al. 2009 [33] and SHARP [56]] are replicated by the same meta-analyses [Rodrigues et al. 2016 [43] and Niu et al. 2016 [42]].

| Highly Cited Study | Replication | PICO General Components | Original conclusion | Effect Measure | Highly Cited ES [95% CI] | Replication ES [95% CI] |
|---|---|---|---|---|---|---|
| Brahmer et al., 2012 [45] | Zhang et al., 2016 [46] | Advanced cancer, Nivolumab, NA, Response | Nivolumab induced tumor regression and prolonged stabilization of disease in advanced cancers | ORR | 0.13 [0.06–0.17] | 0.27 [0.21–0.33] |
| Topalian. et al. 2012 [47] | Tie et al., 2017 [48] | Cancer, Nivolumab, NA, Tumour Response | Nivolumab produced objective responses in cancer regression. | ORR | 0.21 [0.16–0.26] | 0.26 [0.21–0.31] |
| TAXUS-IV [Stone et al., 2004] [49] | Bangalore et al., 2013 [50] | Coronary artery disease, Paclitaxel stent, Metal stent, Revascularization | A paclitaxel-eluting stent reduced the rate of clinical and angiographic restenosis at nine months | RR | 0.39 [0.26–0.59] | 0.66 [0.59–0.74] |
| SYNTAX [Serruys et al., 2009] [51] | Ali et al., 2018 [52] | Severe coronary artery disease, Percutaneous coronary intervention [PCI], Coronary-artery bypass grafting [CABG], Cardiovascular and cerebrovascular events | CABG resulted in lower rates of major adverse cardiac or cerebrovascular events than PCI after 1 year | OR | 1.44 [1.11–1.89] | 1.42 [1.27–1.59] |
| HERA [Piccart-Gebhart et al., 2005] [53] | Genuino et al., 2019 [54] | HER2-positive breast cancer after adjuvant chemotherapy, Trastuzumab, Observation, Progression | Trastuzumab after adjuvant chemotherapy improved disease-free survival in HER2-positive breast cancer | HR | 0.54 [0.43–0.67] | 0.65 [0.55–0.75] |
| CATIE [Lieberman et al., 2005] [27] | Soares-Weiser et al., 2013 [55] | Schizophrenia, Olanzapine, Quetiapine, Treatment discontinuation | Time to discontinuation of treatment for any cause was longer for olanzapine than for quetiapine | HR | 0.63 [0.52–0.76] | 0.68 [0.56–0.83] |
| SHARP [Llovet et al., 2008] [56] | Niu et al., 2016 [42] | Advanced hepatocellular carcinoma, Sorafenib, Placebo, Survival | Survival was longer with sorafenib than with placebo | HR | 0.69 [0.55–0.87] | 0.69 [0.60–0.79] |
| ERSPC [Schröder et al., 2009] [29] | Ilic et al., 2018 [57] | Middle- to old-age men, PSA screening, Control (no screening), Prostate cancer death | Periodic PSA-based screening reduced the rate of death from prostate cancer after a median follow-up of 9 years | IRR | 0.80 [0.65–0.98] | 0.96 [0.85–1.08] |
| PROFILE 1007 [Shaw et al., 2013] [58] | Elliott et al., 2020 [59] | Advanced ALK-positive lung cancer, Crizotinib, Chemotherapy, Progression | Crizotinib was superior to standard chemotherapy in preventing cancer progression | HR | 0.49 [0.37–0.64] | 0.46 [0.39–0.54] |
| ACCORD [Action to Control Cardiovascular Risk in Diabetes Study Group et al., 2008] [60] | Fang et al., 2016 [61] | Type 2 diabetes, Intensive glucose control, Standard therapy, Cardiovascular events or cardiovascular death | As compared with standard therapy, intensive therapy did not reduce major cardiovascular events after a mean follow-up of 3.5 years | RR | 0.95 [0.82–1.09] | 0.92 [0.85–1.00] |
| Cheng et al., 2009 [33] | Niu et al., 2016 [42] | Advanced hepatocellular carcinoma, Sorafenib, Placebo, Survival | Sorafenib increased survival in advanced hepatocellular carcinoma when compared with placebo | HR | 0.68 [0.50–0.93] | 0.69 [0.60–0.79] |
| EXTEND-IA [Campbell et al., 2015] [26] | Goyal et al., 2016 [62] | Ischemic stroke, Endovascular thrombectomy + Alteplase, Alteplase, Disability | Early thrombectomy with the Solitaire FR stent retriever improved disability in ischemic stroke as compared with alteplase alone | OR | 6 [2–18] | 4.04 [2.75–5.93] |
| PARTNER A [Smith et al., 2011] [35] | US Core Valve Study [Adams et al.] [38] | Aortic stenosis, Transcatheter aortic-valve implantation (TAVI), Surgical replacement, Mortality | Transcatheter and surgical procedures for aortic valve replacement were associated with similar rates of survival at 1 year in high-risk patients | RR | 0.98 [0.75–1.26] | 0.73 [0.54–0.98] |
| MR CLEAN [Berkhemer et al., 2015] [63] | Rodrigues et al., 2016 [43] | Acute ischemic stroke, Intraarterial treatment + Usual care, Usual care, Disability | Intraarterial treatment administered within 6 hours after stroke onset was effective in reducing disability assessed at 90 days post-intervention. | RR | 1.73 [1.27–2.35] | 1.37 [1.14–1.64] |

*(Continued)*

**Table 3.** (Continued)

| Highly Cited Study | Replication | PICO General Components | Original conclusion | Effect Measure | Highly Cited ES [95% CI] | Replication ES [95% CI] |
|---|---|---|---|---|---|---|
| ECASS III [Hacke et al., 2008] [64] | Wardlaw et al., 2012 [65] | Acute ischemic stroke, Alteplase, Placebo, Disability | Intravenous alteplase improved disability in patients with acute ischemic stroke assessed 90 days after the intervention | OR | 1.34 [1.02–1.76] | 1.29 [1.16–1.43] |
| EURTAC [Rosell et al., 2012] [30] | Zhao et al., 2019 [44] | Advanced EGFR mutation-positive non-small-cell lung cancer, Erlotinib, Standard chemotherapy, Progression | Erlotinib increased progression-free survival when compared to standard chemotherapy in Asian and European patients with advanced EGFR mutation-positive non-small-cell-lung cancer | HR | 0.37 [0.25–0.54] | 0.23 [0.17–0.30] |
| ESCAPE [Goyal et al., 2015] [66] | Rodrigues et al., 2016 [43] | Ischemic stroke, Standard care + Endovascular treatment, Standard care, Disability | Rapid endovascular treatment [thrombectomy] improved functional outcomes in up to 12 hours after symptom onset | RR | 1.86 [1.39–2.47] | 1.37 [1.14–1.64] |
| NEJSG [Maemondo et al., 2010] [32] | IPASS [Mok et al., 2009] [31] | Non-small-cell lung cancer with mutated EGFR, Gefitinib, Carboplatin–paclitaxel, Progression | First-line gefitinib improved progression-free survival as compared with standard chemotherapy in patients with non-small-cell lung cancer with EGFR mutations. | HR | 0.32 [0.22–0.41] | 0.48 [0.36–0.64] |
| Hamid et al., 2013 [41] | Pyo et al., 2017 [67] | Melanoma, Lambrolizumab, NA, Tumor response | In advanced melanoma, treatment with lambrolizumab resulted in a high rate of sustained tumor regression | ORR | 0.38 [0.25–0.44] | 0.29 [0.26–0.32] |
| KEYNOTE-024 [Reck et al., 2016] [68] | KEYNOTE-042 [Mok et al., 2019] [69] | PD-L1-positive non-small-cell lung cancer, Pembrolizumab, Chemotherapy, Survival | Pembrolizumab led to longer overall survival than platinum-based chemotherapy for PD-L1-positive non-small-cell lung cancer | HR | 0.50 [0.37–0.68] | 1.07 [0.94–1.21] |
| KEYNOTE-001 [Garon et al., 2015] [70] | KEYNOTE-010 [Herbst et al., 2016] [71] | Non–small-cell lung cancer, Pembrolizumab, NA, Tumor Response | Pembrolizumab showed antitumor activity and led to objective responses in patients with advanced non–small-cell lung cancer | ORR | 0.19 [0.16–0.23] | 0.25 [0.22–0.29] |
| Wolchock et al., 2013 [72] | Checkmate 067 [Larkin et al., 2015] [25] | Advanced Melanoma, Nivolumab + Ipilimumab, NA, Response | Concurrent therapy with nivolumab and ipilimumab led to tumor regression in a substantial proportion of patients | ORR | 0.40 [0.27–0.55] | 0.44 [0.38–0.49] |
| Flaherty et al., 2010 [73] | BRIM-3 [Chapman et al., 2011] [24] | Metastatic melanoma with activated BRAF, PLX4032 [Vemurafenib], NA, Response | Treatment of metastatic melanoma carrying the V600E BRAF mutation with PLX4032 [Vemurafenib] resulted in tumor regression | ORR | 0.81 [0.63–0.93] | 0.48 [0.42–0.55] |
| Fong et al., 2009 [74] | Kaufman et al., 2015 [75] | BRCA1- or BRCA2-mutated resistant cancers, Olaparib (AZD2281), NA, Tumor Response | Olaparib had antitumor activity and led to objective responses in cancers associated with BRCA1 or BRCA2 mutations | ORR | 0.47 [0.24–0.71] | 0.26 [0.21–0.32] |

No evidence of effect size inflation was observed in our sample (Fig 3 and Table 5), with the average ratio between the effect sizes of replications and those of highly cited studies approaching 1 when publication order was considered. Inflation increased slightly when effect sizes were coined (i.e., when the percentage of non-responders was used as the outcome) and when publication order was not taken into account (i.e., when highly cited studies were always

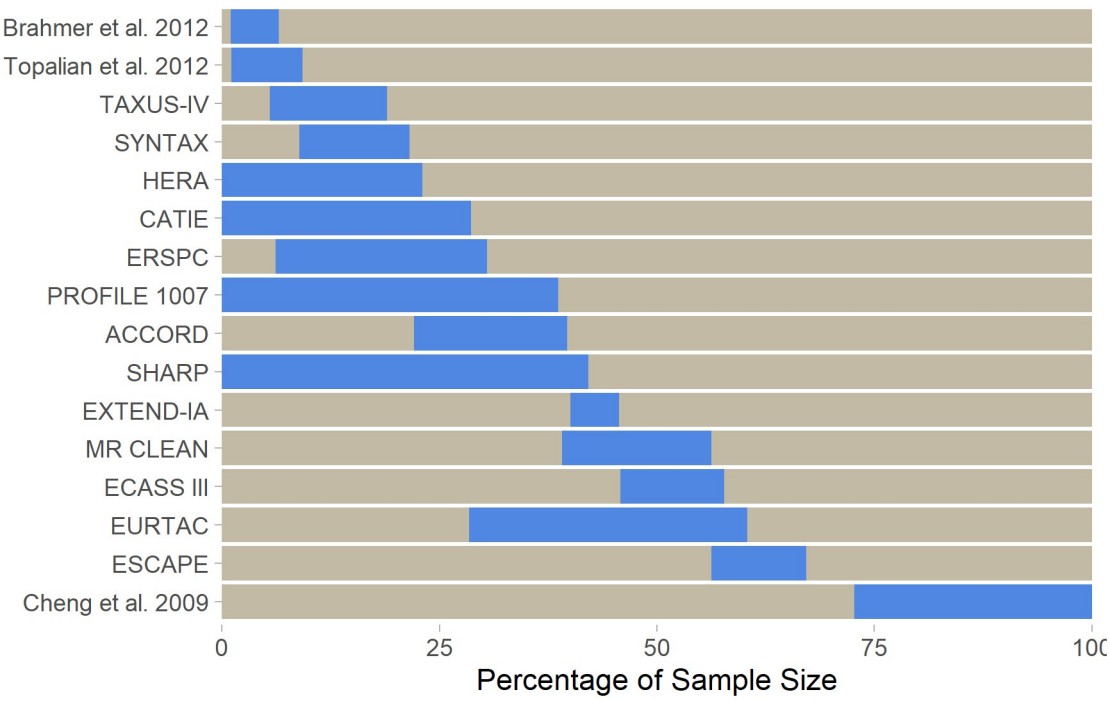

**Fig 2. Relative contribution of highly cited studies to the sample size of replication meta-analyses.** For studies with meta-analyses as replications, blue bars show the fraction of the total sample size that corresponds to the highly cited study, while gray bars represent the rest of the sample size. Fractions are arranged in temporal order from left to right, so that gray bars to the left of the blue ones represent studies that came before the highly cited one. The meta-analysis [67] replicating Hamid et al., 2013 [41] is not shown because the highly cited study is not included in it due to the lack of a control group.

considered as the reference), but remained low on average and did not reach statistical significance in any of our analyses. This picture did not change when only fully independent replications were used to estimate inflation (S1 Fig and S3 Table).

Potential predictors of replicability are shown in Table 6. Although this analysis was planned in the protocol, it has low statistical power due to the low number of contradicted studies in our sample. The same analysis using only fully independent replications is shown on S4 Table. In both analyses, low power prevents us from drawing any definite conclusions on predictors of replicability.

**Table 4. Rate of successful replication as measured by different criteria.** Results are shown as the percentage of studies with eligible replications in which the original result was replicated by each of 5 different criteria. For the statistical significance criterion, we excluded 7 phase 1 trials that did not report p-values and 2 studies with non-significant results. For these studies, the aggregate criterion (CI overlap + statistical significance) only considers CI overlap. The 2 negative studies are included in the "statistical significance (including negative studies)" criterion as a sensitivity analysis.

| Criterion | Total | Replicated | % Replicated [95% CI] |
|---|---|---|---|
| Statistical significance | 15 | 13 | 87% [62, 96] |
| 95% CI overlap | 24 | 21 | 88% [69, 96] |
| 95% CI overlap and statistical significance | 24 | 20 | 83% [64, 93] |
| Statistical significance [including negative studies] | 17 | 13 | 76% [56; 96] |
| Replication estimate within highly cited study's 95% CI | 24 | 15 | 62% [43, 79] |
| Highly cited study estimate within replication 95% CI | 24 | 9 | 38% [21, 57] |

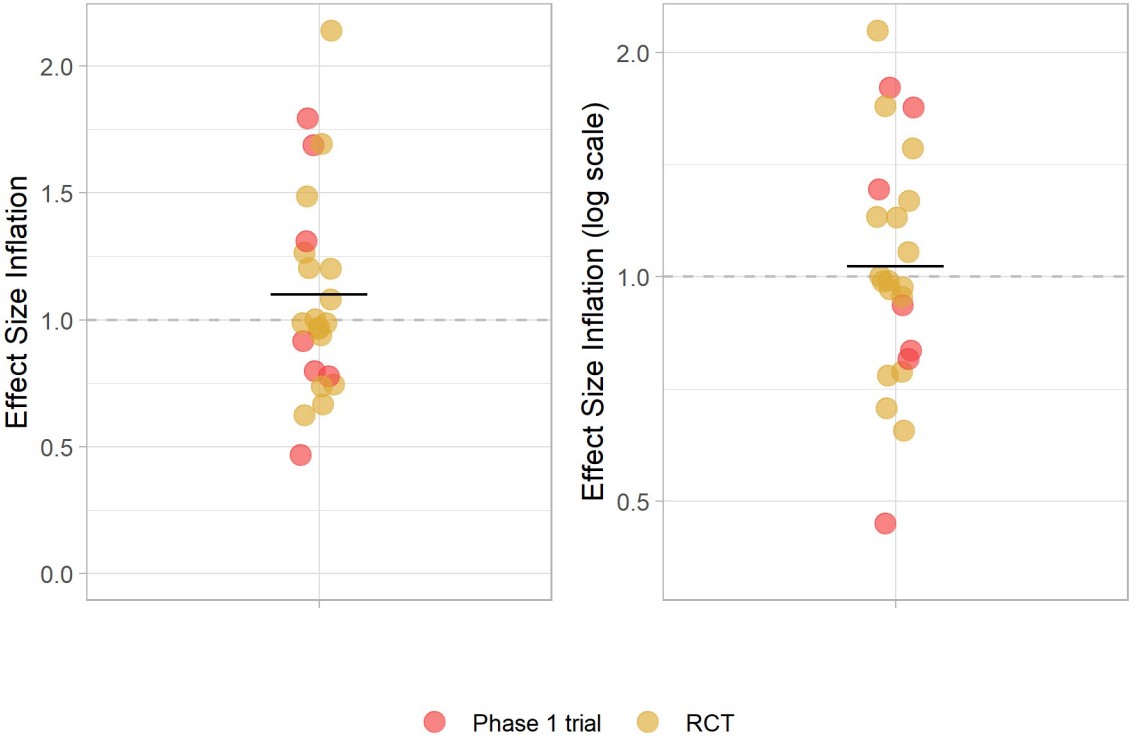

**Fig 3. Effect size inflation.** Effect size inflation was calculated for each study pair considering the effect sizes of highly cited studies and their respective replications. For unfavorable outcomes (e.g., death, tumor response), inflation was calculated as the ratio between the replication and highly cited point estimates. For favorable outcomes (e.g., neurologic improvement), it was calculated as the ratio between the highly cited and the replication point estimates. When the highly cited study came after most or all of the replication studies, we considered the highly cited study as the replication and inverted the ratio. Left panel shows the results in a linear scale (in which distribution is expected to be skewed upwards even in the absence of inflation because of the nature of ratios), while right panel shows them in a log-scale (in which values should be distributed symmetrically around 1 in the absence of inflation). Lines indicate the mean of the plotted values (which in the left panel differs from that on Table 5, calculated on the basis of log-transformed values). Colors indicate Phase 1 trials (red) and RCTs (orange).

## Discussion

### Replicability of highly cited clinical research

The replicability of highly cited clinical studies in our sample was high, with an 83% replication rate when considering our primary outcome of overlapping CIs along with statistical

**Table 5. Analyses of effect size inflation.** Analyses to detect evidence of effect size inflation using different ways to handle favorable/unfavorable outcomes and meta-analyses with most of the sample size preceding the highly cited study. In "Publication order", we did not invert the ratio if a replication had most or all of its sample size published before the highly cited study. In "effect coining", we used unfavorable outcomes for all effect measures, thus avoiding the need to reverse ratios for favorable ones. The bottom row combines both approaches. P-values refer to a one-sample Student's t test against a theoretical mean of 0 for the log-transformed inflation ratios for each study pair. Values for means and 95% CIs were calculated for these log-transformed ratios and exponentiated back to the original scale. Inflation point estimates were manually replicated with the results of the R code output (https://osf.io/9hmx4) and this analysis can be found at https://osf.io/qnfgh.

| Analysis | All | p-value | Phase 1 trials | RCTs |
|---|---|---|---|---|
| Main analysis | 1.03 [0.88–1.21] | 0.68 | 1.01 [0.64–1.57] | 1.04 [0.88–1.23] |
| Publication order | 1.10 [0.95–1.28] | 0.20 | 1.01 [0.64–1.57] | 1.14 [0.98–1.33] |
| Effect coining | 1.07 [0.93–1.24] | 0.34 | 1.17 [0.80–1.71] | 1.03 [0.88–1.22] |
| Publication order + coining | 1.14 [0.99–1.30] | 0.07 | 1.17 [0.80–1.71] | 1.12 [0.96–1.31] |

**Table 6. Predictors of replication success.** Table shows comparisons of potential predictors of replicability between replicated and contradicted studies using different definitions of replication success. P-values are shown for a Mann-Whitney test (for continuous variables) or Fisher's exact test (for categorical variables) comparing replicated and contradicted studies. For the significance criterion, only 15 studies (i.e., initially positive RCTs) were considered. Study design is not included as a predictor of statistical significance, as phase 1 trials did not use significance testing. Sample size for each group varies according to the specific criteria: statistical significance: 13 replicated, 2 contradicted; CI overlap: 21 replicated, 3 contradicted; both criteria: 20 replicated, 4 contradicted. IQR = interquartile range.

| Predictor | Criteria | Median [IQR] Replicated | Median [IQR] Contradicted | p-value |
|---|---|---|---|---|
| Citations/year of the highly cited study | Significance [p < 0.05] | 292 [250–454] | 441 [338–543] | 0.69 |
| | CI overlap | 329 [250–454] | 494 [368–570] | 0.35 |
| | Significance [p < 0.05] & CI overlap | 337 [270–459] | 368 [241–532] | 0.79 |
| p-value of the highly cited study | Significance [p < 0.05] | $3 \times 10^{-5}$ [$7 \times 10^{-7}$ – $2 \times 10^{-3}$] | 0.02 [$8 \times 10^{-3}$–0.02] | 0.48 |
| | CI overlap | $1 \times 10^{-3}$ [$4 \times 10^{-6}$–0.02] | $1 \times 10^{-5}$ [$1 \times 10^{-5}$ – $1 \times 10^{-5}$] | 0.82 |
| | Significance [p < 0.05] & CI overlap | $5 \times 10^{-4}$ [$3 \times 10^{-6}$–0.01] | 0.02 [$8 \times 10^{-3}$–0.02] | 0.72 |
| Sample size of the highly cited study | Significance [p < 0.05] | 500 [226–821] | 91152 [45729–136576] | 0.57 |
| | CI overlap | 495 [200–821] | 207 [131–256] | 0.17 |
| | Significance [p < 0.05] & CI overlap | 421 [194–730] | 256 [169–45729] | 0.74 |
| **Predictor** | **Criteria** | **# replicated by study design** | **# not replicated by study design** | **p-value** |
| Highly cited study design | CI overlap | Phase 1 trial: 5/7 RCT: 16/17 | Phase 1 trial: 2/7 RCT: 1/17 | 0.19 |
| | Significance [p < 0.05] & CI overlap | Phase 1 trial: 5/7 RCT: 15/17 | Phase 1 trial: 2/7 RCT: 2/17 | 0.55 |
| Type of intervention | Significance [p < 0.05] | Pharmacological: 7/8 Other: 6/7 | Pharmacological: 1/8 Other: 1/7 | 1.00 |
| | CI overlap | Pharmacological: 13/16 Other: 8/8 | Pharmacological: 3/16 Other: 0/8 | 0.53 |
| | Significance [p < 0.05] & CI overlap | Pharmacological: 13/16 Other: 7/8 | Pharmacological: 3/16 Other: 1/8 | 1.00 |

significance in the same direction. Using only fully independent replications (i.e., including meta-analyses only when highly-cited studies were removed) led to a slightly lower but still reasonably high estimate of 79%. Moreover, we did not find evidence of systematic effect size inflation, either for phase 1 trials or for RCTs.

These replication success rates are higher than those found in previous studies of highly cited clinical literature, where rates of 59% in general medicine [11] and 37% in psychiatry [12] were described when both statistical significance and effect size were considered, although the criteria for comparing effect sizes differed in each study. For statistical significance alone (a more homogeneous criterion), the successful replication rate for studies with significant results was 87% in our study, as compared to 79% [11] and 63% [12] in the previous two studies, respectively.

Although the differences in these estimates could be due to changes in the replicability of the published literature, methodological discrepancies between studies should be considered. Ioannidis' study [11], used a broader definition of replication; thus, many article pairs in his sample would not have been considered to have matching PICO components by our criteria. Moreover, among the contradicted or inflated studies in his sample, 4 were cohort studies whose replications were RCTs or meta-analyses of RCTs. Due to the presence of these cohort studies (as well as to the lower frequency of phase 1 studies), median sample size is larger in Ioannidis's sample (median = 1500, IQR = 633–4382) than in ours (median = 332, IQR = 194–730). The author himself acknowledges that it is not always possible to validate exposures as interventions, and other studies comparing observational research with RCTs have used the term "concordance" instead of replicability [76]. If one considers only interventional studies

(i.e., RCTs and case series) in Ioannidis' study [11]–as was the case in our sample–, the replication rate is 67% when considering both significance and effect size, or 87%–exactly the same as ours when including only statistically significant highly cited studies–when considering statistical significance alone.

## Defining replication boundaries

Considering a study as a replication of another inevitably requires establishing the boundary conditions of a claim [10]. We opted to define replications as studies that had matching PICO general components [77]. This led most highly cited studies to be classified as unchallenged, with replications being found for only 27% of our sample [as opposed to 76% in Ioannidis [11], 52% in Tajika et al. [12], and 42% in Niven et al. [13], in which criteria were less stringent]. Many studies that could have been considered replications by looser criteria were thus excluded.

Even though we were more conservative than previous studies in defining replication boundaries, our study pairs were still not perfect replicas of each other. In some cases, definitions for clinical conditions were very broad, such as "cancer" in Brahmer et al. 2012 [45] and Topalian et al. 2012 [47], meaning that replication samples could potentially be quite distinct from the original one. Our definition also allowed for variation in methodological details between studies, such as the treatment protocol and follow-up length, as long as the PICO general components were kept constant. These discrepancies thus remain as possible explanations for contradictions between results. Heterogeneity between study populations and interventions presents challenges to studying replicability in clinical research, and methodological differences between the original study and replications seem to be common in previous studies as well [13].

## Contradicted studies

Regarding our primary outcome, two phase 1 trials and two RCTs were classified as contradicted. Both phase 1 trials had CIs that did not overlap with those of the replication–in one case, the effect was larger in the original study, while in the other it was larger in the replication. As phase 1 trials typically have small sample sizes and are likely to be more prone to publication/citation bias (i.e., a negative phase 1 trial is unlikely to become highly cited), their replicability is expected to be lower than that of RCTs. Nevertheless, the majority of phase 1 trials in our sample were successfully replicated, although replication criteria were less stringent for these studies as (a) they were not subject to the statistical significance criteria and (b) CIs for their effect sizes were broader. Still, it's worth noting that all RCTs replicating phase 1 trials in our sample showed a statistically significant benefit of the intervention when compared to a control group.

Concerning RCTs, the two replication failures for initially positive studies were observed for ERSPC [29] (a large prostate cancer screening trial) and KEYNOTE-024 [68] (a trial of the checkpoint inhibitor pembrolizumab in lung cancer). ERSPC [29] was contradicted because it showed a statistically significant effect [p = 0.03] while the meta-analysis did not reach significance, although effect sizes do overlap. Some methodological issues have been proposed to account for this discrepancy, such as the large degree of control group contamination in the PLCO trial [78] and the lower screening intensity in the CAP trial [79], two negative studies that account for most of the weight in the replication meta-analysis.

KEYNOTE-024 [68], meanwhile, was considered contradicted by our replication criteria, which specifically addressed the primary outcome in the highly cited study–in this case, progression-free survival. Nevertheless, both the original study and the replication [KEYNOTE-

042] [69] showed an overall survival benefit, and the lack of effect on progression in the replication seems to be due to differences in the early stage of the trial, even though the intervention group fared better on the long run. Thus, the result was replicated when the more clinically relevant outcome of survival is considered, and the lack of replication for our chosen outcome appears to be a statistical accident.

Of note, we did not use the statistical significance criterion to evaluate studies with non-significant results, as lack of statistical significance should not be taken as evidence of equivalence between treatments (an erroneous assumption that sometimes occurs in the literature) [80]. Accordingly, of the two non-significant studies in our sample, one of them–the ACCORD trial [60]–had a replication with a significant result–in this case, a large meta-analysis that reached marginal significance (p = 0.04). Nevertheless, other meta-analyses arrived at different conclusions [61, 81–84]. Effect sizes were similar between studies, suggesting that lack of agreement in this criterion was a consequence of lower statistical power in the highly cited study. PART-NER A [35], meanwhile, was a non-inferiority study that showed similar 1-year outcomes with transcatheter and surgical aortic valve replacement. Its replication [38] found a better outcome in the transcatheter group, with the authors speculating that the contrasting results could be due to the type of prosthesis used or to differences in the patients' risk profile. In both studies, the replication result barely passes the 0.05 threshold, demonstrating that non-replication can arise merely as a matter of statistical fluctuation when significance-based criteria are used to define replication success.

## Effect size inflation

Many studies analyzing replications have found evidence that published effects are systematically inflated, a fact that is expected when statistical significance thresholds are used as a criterion for publication [85]. Nevertheless, strategies to measure effect size inflation vary widely across studies. Ioannidis found initially stronger effects in 7 out of 27 replicated studies, using the criteria of a decrease in risk reduction of at least 50%, or a benefit of shorter duration or limited generalizability in the replication [11]. Tajika et al. reported standardized mean differences of initial studies to be 2.3 times larger than those of replications [12], while Niven et al. [13] reported a mean absolute risk difference of 16% between original studies and replications. Similar evidence of effect size inflation has also been found both in systematic replication initiatives [5, 8] and meta-analyses of effect sizes over time [86, 87].

Contrary to these studies, we found little evidence of systematic effect size inflation in the highly cited clinical literature between 2004 and 2018. This suggests that publication/citation bias might be more limited in our sample than in other fields of research. That said, our ability to detect it in our primary analysis could have been reduced by the use of meta-analyses as replications, as the replication sample included the highly cited study, as well as some studies that came before it. Nevertheless, our supplementary analysis removing the primary studies from the meta-analyses found very similar inflation estimates, suggesting that this was not a major issue.

Another limitation is that using ratios for measuring effect size inflation leads to variation in estimates depending on the outcome used. If nonresponse rates or response odds were used instead of ORRs for phase 1 trials, for example, estimates of inflation increased to 17% or 18%, respectively–although CIs were still wide and compatible with absence of systematic inflation. One can also make the case that relative differences in effect sizes may be less relevant than absolute ones for clinical practice. Nevertheless, the fact that different outcome measures were used across studies makes absolute differences non-commensurable and prevents us from analyzing the sample as a whole in this manner.

Although evidence for systematic inflation was limited, this does not mean that initially stronger effects were not found in some studies, as in the case of KEYNOTE-024 [69], in which a risk reduction in progression disappeared in the replication, and TAXUS-IV [49], in which relative risk in the treated group increased from 0.39 to 0.66. Nevertheless, the fact that increases in effect size were found in other studies–such as Brahmer et al., 2012 [45], in which the ORR doubled from 13% to 27%,–suggests that some or most of these discrepancies can be explained by statistical fluctuation, and that systematic bias in favor of positive studies is smaller in this literature than in other fields.

## Replication criteria

Different replication criteria complement each other by capturing distinct aspects of replicability [6, 8]. Statistical significance alone does not distinguish between magnitude and precision, and thus says little about how two effects compare directly [88]. Comparing effect sizes, meanwhile, avoids emphasis on statistical thresholds [8], but may lead to studies with different conclusions be considered as successful replications of each other. For this reason, our primary outcome was a combination of statistical significance and CI overlap of effect sizes.

Highly cited studies were expected to predominantly present statistically significant results for their primary outcomes, as this literature is enriched in studies with high power and high prior probabilities [89]. Most replications also yielded significant results, something that would be expected if the primary findings represent true effects. In fact, considering that the replication rate for this criterion was 87% for initially positive studies, even replication failures could represent vibration around statistical thresholds (as might have been the case for the KEYNOTE-024 replication [68], for example), as studies in clinical medicine are often powered around that level.

Replicability based on CI overlap was similarly high [88%], although this is a rather loose threshold for effect size similarity: absence of CI overlap for two identical effects is expected to occur by chance alone in around 0.6% of cases [90], and this high bar for type 1 error comes at the cost of lower statistical power to detect differences in effect sizes. A more stringent criterion of having the replication effect size included in the original CI led to a lower but still reasonable [62%] replication rate, despite not considering the potential variability in replication estimates.

Inclusion of the highly cited study's point estimate in the replication CI was less frequent [38%], but this is an overly strict criterion, especially when replications have sample sizes that are much larger than the original studies. Calculating prediction intervals for the original effect given the replication sample size would likely represent the fairest way to assess replication of effect sizes [91], but this was not always possible for all cases based on the available data.

## General limitations

Defining what constitutes a replication is not trivial: even though we followed a predefined protocol to define PICO components, their abstraction inevitably involves a degree of subjectivity. Moreover, as was the case in previous studies of replications in the published literature [11–13, 87], there was no way to develop a systematic search strategy that was applicable for every study. Because of these factors, our independent searches for replications had low agreement, and a second step was needed to reach consensus. Still, it is possible that our searches could have missed some valid replication candidates.

In at least one case–the ACCORD trial on intensive glucose control [61]–there were candidate replications that reached different results [60, 61, 82–84], and controversy around replications and their methodology are common [92]. Although we used an objective criterion to

define the replication selected for analysis [i.e., sample size], a replication with a different result might have been chosen if we used other criteria. Of note, we did not evaluate risk of bias or methodological quality in replications, opening up the possibility that the largest replication available might not be necessarily the most reliable one.

An important caveat in our analysis is the fact that meta-analyses were considered as replications, even though most of them included the highly cited study. This leads to a degree of circularity in the analysis that could have biased our reproducibility estimate upwards. To deal with this problem, we conducted independent meta-analyses excluding the highly cited study in order to turn them into truly independent replications. Interestingly, this did not lead to major changes in our replication rates, and actually led to lower estimates of effect size inflation. This confirms our impression that highly cited clinical literature seems to be generally replicable, and that these studies' effect sizes are not systematically higher than those of other studies on the same topic.

Even after including meta-analyses, our stringent criteria to consider studies as having matching PICO components left us with a small sample, and the high replicability rate led to an even lower number of contradicted studies. Thus, our analysis was markedly with low statistical power to detect predictors of replicability. Even though none of our predictors reached statistical significance, it seems likely that factors such as lower p values or higher sample sizes would be associated with a higher replicability rate, as found in other areas [8, 93], had a larger sample been available.

As a final limitation, we relied on replications that were published in the literature. As the existence and publication of these replications are subject both to the interest of researchers to perform them and to that of editors and reviewers to publish them, the approach in our study is not directly comparable to the systematic replication attempts that have been performed in other areas [1–8]. This is particularly important given that the majority of highly cited studies in our sample had no available replications according to our criteria–thus, selectiveness in performing or publishing replications may have biased our replicability rates. Theoretical discussions propose that replication value is highest for articles that receive a large amount of attention (as measured by citations, for instance) while still possessing a high degree of effect uncertainty [94]; nevertheless, it is unclear whether these factors are actually taken into account by clinical researchers building upon previous work.

It is also possible that successfully replicated studies receive more citations in the long run, biasing the reproducibility rate of highly cited studies upwards. Nevertheless, existing evidence suggests that this might not be the case: a citation analysis of a sample of psychology, economics and social science articles included in replication projects showed that non-replicated studies actually received more citations than replicated ones [95], while other studies found no significant differences [93, 96]. Case studies also suggest that the impact of claims that fail to replicate on subsequent citations is modest, both in clinical medicine [97] and in psychology [98]. Nevertheless, due to the very selective nature of our sample, the possibility that different biases may occur in the highly cited clinical literature cannot be excluded.

## Conclusions

Despite the high rate of unchallenged studies, we found the replicability rate of the highly cited clinical literature between 2004 and 2018 to be higher than previously estimated, with little evidence of effect size inflation. These numbers are valid for a narrow, very influential subsample of articles, and cannot be generalized to medical research at large. Moreover, they could be biased by factors that influence the selection of studies for replication or the citation patterns of replicated studies. Nevertheless, our findings run counter to the assertion that there is a

widespread reproducibility crisis in science, and suggest that this may not be the case for every scientific field.

The higher replication rate found in our study when compared to earlier samples of the clinical literature could also be taken as a sign of improvement over time; nevertheless, this conclusion is tentative at best, as differences in methodology (such as the definition of effect size inflation) and samples (such as the frequency of different study designs) do not warrant direct comparisons between studies.

Further research is warranted to examine whether the high replicability of highly cited clinical research is related to particular research practices that are not as widely used in other areas of biomedical science, such as randomization, blinding or prospective protocol registration [14, 16, 99–102]. If such links can be reliably established, they could be used to inform attempts to improve replicability in different research fields.

## Supporting information

**S1 Table. Results from reanalyzed meta-analyses with and without the highly cited study.** Table shows highly cited studies with their effect measures, the % of sample size they account for in the replication meta-analyses, the published results of these meta-analyses, and those of their reanalyses before and after removing the highly cited study. Out of 16 meta-analyses, we were able to reanalyze 11: EXTEND-IA (12) is an individual patient data meta-analysis, whereas the remaining 3 are network meta-analyses with no direct comparisons. One network meta-analysis (17) replicates two highly cited studies ((13) and (16)), making it 4 network meta-analyses that were not possible to conduct independent replications. One network meta-analysis (18) had direct comparisons in its supplementary materials, making it possible to conduct the independent analysis. Differences between the effect sizes of published and reanalyzed meta-analyses using the same studies occur due to changes in meta-analytical methods and software, but are generally small.
(DOCX)

**S2 Table. Replication rates using fully independent replications only.** Rates consider only independent primary studies (i.e. RCTs, phase II trials) and meta-analyses that do not include the highly cited studies. Meta-analyses that could not be reanalyzed were excluded from the analysis. Otherwise, results are displayed in the same way as in Table 4.
(DOCX)

**S3 Table. Effect size inflation using fully independent replications only.** Rates consider only independent primary studies (i.e. RCTs, phase II trials) and meta-analyses that do not include the highly cited studies. Meta-analyses that could not be reanalyzed were excluded from the analysis. Otherwise, results are displayed in the same way as in Table 5.
(DOCX)

**S4 Table. Predictors of replicability using fully independent replications only.** Rates consider only independent primary studies (i.e., RCTs, phase II trials) and meta-analyses that do not include the highly cited studies. Meta-analyses that could not be reanalyzed for this purpose were excluded from the analysis. Otherwise, results are displayed in the same way as in Table 6. Sample size for each group varies according to the specific criteria: statistical significance: 8 replicated, 2 contradicted; confidence interval overlap: 16 replicated, 3 contradicted; both criteria: 15 replicated, 4 contradicted.
(DOCX)

**S1 Fig. Inflation of effects using fully independent replications only.** Rates consider only independent primary studies [i.e. RCTs, phase II trials] and meta-analyses that do not include the highly cited studies. Meta-analyses that could not be reanalyzed for this purpose were excluded from the analysis. Otherwise, results are displayed in the same way as in Fig 3. Lines indicate the mean of the plotted values (which in the left panel differs from that on S3 Table, calculated on the basis of log-transformed values).
(TIFF)

## Acknowledgments

An abstract for this study has been published in *BMJ Evidence-Based Medicine* (http://dx.doi.org/10.1136/bmjebm-2022-PODabstracts.105) as a product of the EBM Live 2022 Conference.

## Author Contributions

**Conceptualization:** Gabriel Gonçalves da Costa, Kleber Neves, Olavo Amaral.

**Data curation:** Gabriel Gonçalves da Costa, Kleber Neves.

**Formal analysis:** Gabriel Gonçalves da Costa, Kleber Neves.

**Investigation:** Gabriel Gonçalves da Costa, Kleber Neves, Olavo Amaral.

**Methodology:** Gabriel Gonçalves da Costa, Kleber Neves, Olavo Amaral.

**Project administration:** Gabriel Gonçalves da Costa, Olavo Amaral.

**Resources:** Gabriel Gonçalves da Costa, Olavo Amaral.

**Software:** Gabriel Gonçalves da Costa, Kleber Neves.

**Supervision:** Gabriel Gonçalves da Costa, Olavo Amaral.

**Validation:** Gabriel Gonçalves da Costa, Olavo Amaral.

**Visualization:** Gabriel Gonçalves da Costa, Olavo Amaral.

**Writing – original draft:** Gabriel Gonçalves da Costa, Olavo Amaral.

**Writing – review & editing:** Gabriel Gonçalves da Costa, Kleber Neves, Olavo Amaral.

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
