## [Decision Letter · Decision Letter 0]

6 Feb 2024

PONE-D-23-26404Estimating the replicability of highly cited clinical research (2004-2018)PLOS ONE

Dear Dr. da Costa,

Thank you for submitting your manuscript to PLOS ONE. After careful consideration, we feel that it has merit but does not fully meet PLOS ONE’s publication criteria as it currently stands. Therefore, we invite you to submit a revised version of the manuscript that addresses the points raised during the review process.

We look forward to receiving your revised manuscript.

Kind regards,

Tim Mathes

Academic Editor

PLOS ONE

Journal Requirements:

2. Please be informed that funding information should not appear in the Acknowledgments section or other areas of your manuscript. We will only publish funding information present in the Funding Statement section of the online submission form. Please remove any funding-related text from the manuscript.

3. Please remove your figures from within your manuscript file, leaving only the individual TIFF/EPS image files, uploaded separately. These will be automatically included in the reviewers’ PDF.

Reviewers' comments:

Reviewer's Responses to Questions

**Comments to the Author**

1. Is the manuscript technically sound, and do the data support the conclusions?

Reviewer #1: Yes

Reviewer #2: Yes

2. Has the statistical analysis been performed appropriately and rigorously? 

Reviewer #1: Yes

Reviewer #2: Yes

3. Have the authors made all data underlying the findings in their manuscript fully available?

Reviewer #1: Yes

Reviewer #2: Yes

4. Is the manuscript presented in an intelligible fashion and written in standard English?

Reviewer #1: Yes

Reviewer #2: Yes

5. Review Comments to the Author

Reviewer #1: Review of Estimating the replicability of highly cited clinical research (2004-2018)

This paper examines the replicability of highly cited (>2000 citations) clinical research over the period 2004-2018. In so doing, the paper attempts to provide evidence for or against prior work that has suggested that highly cited articles are often unexpectedly contradicted or found to have inflated effects in subsequent research. The replicability of papers in their sample are evaluated by two criteria (with some reasonable exceptions) by the presence of a statistically significant effect (p <0.05) in the same direction or a finding within the confidence interval (CIs) of the predictor of the original study.

Methodologically, the authors do not do original replications. Rather, they cleverly search for “replications” by other authors that were published in PubMed, or as part of meta-analyzes, and had a reasonable PICO alignment with the original study (Population, Intervention, Comparator and Outcome). They find that in the 89 highly cited they located in the literature, 24 had valid replications (17 meta-analyses and 7 primary studies), of which 21 (88%) had effect sizes with overlapping CIs. Of 15 highly cited studies with a statistically significant difference in the primary outcome, 13 (87%) had a significant effect in the replication as well. When both criteria were considered together, the replicability rate in our sample was of 20 out of 24 (83%). No evidence for inflation was found.

The authors also compare the characteristics of replicating and non-replicating paper in an effort to provide descriptive data on how replicating and non-replicating papers may differ such as citation per year and effect size. Table 3 is a useful qualitative analysis of the papers studied that adds rich detail and context that few replication studies do. The writing is tight and the methodological work does its best to deal with sampling problems and the use of meta-analysis data.

Bottom line, the authors find there was little evidence of inflation and the replicability rate of highly cited papers was higher than seen in previous studies.

I think the paper should be published but conditional on the following changes being made.

1. Report data limitations of the study up front. While it is admirable that the authors are repurposing the replications by other authors to cost-effectively examine the replication rate of highly cited papers, the authors must clearly state that the results must be cautiously interpreted and are preliminary. These highly cited paper were likely put through replication tests by other researchers for reasons that do not generalize (e.g., the papers are especially surprising, have a large or special population sample, present the first of its kind clinical treatment, etc. The sample size of papers is small. Consequently, the authors must clearly report to readers that while their results show higher levels of replicability than previous work, the results rest on methodological and data constraints that have unknown sample selection and sample size biases that limit generalization beyond the papers in the study and that future research is needed before firm conclusions can be drawn.

2. Bring the study up to the literature. Currently, the paper lacks mention of work that makes the paper feel behind the literature. I would suggest updating the paper in several ways. First, the paper appears to ignore the work that has already shown that replicating and non-replicating papers diffuse through the over the first 5 year literature at indistinguishable rates 1,2. Second, in regards to arguments supporting your sampling issue, I would cite 3,4. Finally, a paper that looked at highly cited papers in psychology over a 20 year period showed that highly cited paper replicate at a significantly higher frequency that papers with low citations, bolstering your finding of a higher rate of replication than found in earlier studies 5.

1. Yang Y, Youyou W, Uzzi B. Estimating the deep replicability of scientific findings using human and artificial intelligence. Proceedings of the National Academy of Sciences. 2020;117(20):10762-10768.

2. Hardwicke TE, Szűcs D, Thibault RT, et al. Citation patterns following a strongly contradictory replication result: Four case studies from psychology. Advances in Methods and Practices in Psychological Science. 2021;4(3):25152459211040837.

3. Isager PM, Van Aert R, Bahník Š, et al. Deciding what to replicate: A decision model for replication study selection under resource and knowledge constraints. Psychological Methods. 2021;

4. Cohn A, Fehr E, Maréchal MA. Selective participation may undermine replication attempts. Nature. 2019;575(7782):E1-E2.

5. Youyou W, Yang Y, Uzzi B. A discipline-wide investigation of the replicability of Psychology papers over the past two decades. Proc Natl Acad Sci. 2023;120((6)):e2208863120. doi:https://doi.org/10.1073/pnas.2208863120

Reviewer #2: In the context of the ‘reproducibility crisis’, it is highly important to examine how robust, reliable and replicable clinical trial results are. Furthermore, it also is necessary to examine how replicable the replicability analyses themselves are. In this vein, the present manuscript is a valuable addition to previous studies.

The main issue of the current manuscript is the low rate of replication with matching studies being found for only 27% of primary studies. As only 24 study pairs could be analysed, the results come with some uncertainty. A second problem is the fact, that many studies were replicated by metaanalyses, which included the index study and thus cannot be considered as a fully independent replication. The authors nevertheless address these two points well, so the present analysis is a very useful addition to the existing body of literature. I have only a three minor comments:

1. On page 10, “overall response rates” are mentioned. It appears as if ‘objective response rates’ are meant.

2. On page 26, several reasons are given, why “study pairs were … not perfect replicas of each other”. Perhaps, it would be useful also to take a look at the duration of index and replication study, because it is common that short-term effects wane over time.

3. On page 25, the present analysis is judiciously compared with Ioannidis’ study from 2005. It would be helpful to know the median (and IQR) sample size of the index trials used in Ioannidis’ and those used here, because smaller trials are clearly less reliable than larger ones. It could well be that major journals have become more reluctant to publish smaller trials, even if they show surprisingly positive results. If so, the slightly higher replicability found in the present analysis could be interpreted as a possible improvement in publication policies (or citation patterns).

6. PLOS authors have the option to publish the peer review history of their article (what does this mean?). If published, this will include your full peer review and any attached files.

Reviewer #1: No

Reviewer #2: No

---

## [Author Response · Author response to Decision Letter 0]

13 Jun 2024

Dear Dr. Mathes,

Please find the responses to the reviewer comments below. We hope that our changes to the manuscript will make it acceptable for publication in PloS One.

Yours sincerely,

Gabriel Gonçalves da Costa

Reviewer #1: 

This paper examines the replicability of highly cited (>2000 citations) clinical research over the period 2004-2018. In so doing, the paper attempts to provide evidence for or against prior work that has suggested that highly cited articles are often unexpectedly contradicted or found to have inflated effects in subsequent research. The replicability of papers in their sample are evaluated by two criteria (with some reasonable exceptions) by the presence of a statistically significant effect (p <0.05) in the same direction or a finding within the confidence interval (CIs) of the predictor of the original study.

Methodologically, the authors do not do original replications. Rather, they cleverly search for “replications” by other authors that were published in PubMed, or as part of meta-analyzes, and had a reasonable PICO alignment with the original study (Population, Intervention, Comparator and Outcome). They find that in the 89 highly cited they located in the literature, 24 had valid replications (17 meta-analyses and 7 primary studies), of which 21 (88%) had effect sizes with overlapping CIs. Of 15 highly cited studies with a statistically significant difference in the primary outcome, 13 (87%) had a significant effect in the replication as well. When both criteria were considered together, the replicability rate in our sample was of 20 out of 24 (83%). No evidence for inflation was found.

The authors also compare the characteristics of replicating and non-replicating paper in an effort to provide descriptive data on how replicating and non-replicating papers may differ such as citation per year and effect size. Table 3 is a useful qualitative analysis of the papers studied that adds rich detail and context that few replication studies do. The writing is tight and the methodological work does its best to deal with sampling problems and the use of meta-analysis data.

Bottom line, the authors find there was little evidence of inflation and the replicability rate of highly cited papers was higher than seen in previous studies.

I think the paper should be published but conditional on the following changes being made.

1. Report data limitations of the study up front. While it is admirable that the authors are repurposing the replications by other authors to cost-effectively examine the replication rate of highly cited papers, the authors must clearly state that the results must be cautiously interpreted and are preliminary. These highly cited paper were likely put through replication tests by other researchers for reasons that do not generalize (e.g., the papers are especially surprising, have a large or special population sample, present the first of its kind clinical treatment, etc. The sample size of papers is small. Consequently, the authors must clearly report to readers that while their results show higher levels of replicability than previous work, the results rest on methodological and data constraints that have unknown sample selection and sample size biases that limit generalization beyond the papers in the study and that future research is needed before firm conclusions can be drawn.

We thank the reviewer for the comment, and generally agree with the limitations pointed out above, both in terms of the selectivity (and potential biases) of the sample and the small sample size. These limitations are now pointed out more explicitly in the abstract (page 2), discussion (page 29) and conclusion (page 29), using some of the references suggested in subsequent comments by the reviewer.

2. Bring the study up to the literature. Currently, the paper lacks mention of work that makes the paper feel behind the literature. I would suggest updating the paper in several ways. First, the paper appears to ignore the work that has already shown that replicating and non-replicating papers diffuse through the over the first 5 year literature at indistinguishable rates 1,2.

1. Yang Y, Youyou W, Uzzi B. Estimating the deep replicability of scientific findings using human and artificial intelligence. Proceedings of the National Academy of Sciences. 2020;117(20):10762-10768.

2. Hardwicke TE, Szűcs D, Thibault RT, et al. Citation patterns following a strongly contradictory replication result: Four case studies from psychology. Advances in Methods and Practices in Psychological Science. 2021;4(3):25152459211040837.

Both of these studies are now cited in the discussion (pages 28-29), as well as other studies that have looked at this issue – i.e. Serrano-Garcia & Gneezy, 2021 (https://doi.org/10.1126/sciadv.abd1705), Youyou et al, 2023 (https://doi.org/10.1073/pnas.2208863120) and Tatsioni et al., 2007 (https://doi.org/10.1001/jama.298.21.2517) 

Second, in regards to arguments supporting your sampling issue, I would cite 3,4. 5.

3. Isager PM, Van Aert R, Bahník Š, et al. Deciding what to replicate: A decision model for replication study selection under resource and knowledge constraints. Psychological Methods. 2021;

4. Cohn A, Fehr E, Maréchal MA. Selective participation may undermine replication attempts. Nature. 2019;575(7782):E1-E2.

We have cited Isager et al.’s study as suggested (page 28), although we also note that this is a recent proposal from another field of research, and that it is not clear how much the issues brought up are actually taken into account by clinical researchers building upon previous work. Cohn et al.’s study is added as well on page 27.

Finally, a paper that looked at highly cited papers in psychology over a 20 year period showed that highly cited paper replicate at a significantly higher frequency that papers with low citations, bolstering your finding of a higher rate of replication than found in earlier studies.

5. Youyou W, Yang Y, Uzzi B. A discipline-wide investigation of the replicability of Psychology papers over the past two decades. Proc Natl Acad Sci. 2023;120((6)):e2208863120. doi:https://doi.org/10.1073/pnas.2208863120

As far as we could understand, the finding in this paper is that work from highly cited authors have a higher rate of replication success. The correlation between article citations and replicability was found to be non-significant for empirically replicated papers, and weakly negative for the authors’ metric of predicted replication success. This article is now cited along with references 1 and 2 in the discussion of possible selection bias (page 28).

Reviewer #2:

In the context of the ‘reproducibility crisis’, it is highly important to examine how robust, reliable and replicable clinical trial results are. Furthermore, it also is necessary to examine how replicable the replicability analyses themselves are. In this vein, the present manuscript is a valuable addition to previous studies.

The main issue of the current manuscript is the low rate of replication with matching studies being found for only 27% of primary studies. As only 24 study pairs could be analysed, the results come with some uncertainty. A second problem is the fact, that many studies were replicated by metaanalyses, which included the index study and thus cannot be considered as a fully independent replication. The authors nevertheless address these two points well, so the present analysis is a very useful addition to the existing body of literature. I have only a three minor comments:

1. On page 10, “overall response rates” are mentioned. It appears as if ‘objective response rates’ are meant.

We thank the reviewer for pointing out this mistake, which has been corrected in the revised version (page 9). We noted the same mistake in page 10 and corrected it as well.

2. On page 26, several reasons are given, why “study pairs were … not perfect replicas of each other”. Perhaps, it would be useful also to take a look at the duration of index and replication study, because it is common that short-term effects wane over time.

This is an interesting suggestion, which led us to try to systematically compare follow-up duration between initial studies and replications. However, this turned out to be much more complicated than we thought. Many papers are based on survival analyses (particularly applied to progression-free survival in cancer studies), in which the duration of the study (measured as the median or maximum follow-up) is inevitably linked to the treatment’s efficacy: thus, longer studies by definition correlate with larger treatment effects. Moreover, many of the replications are meta-analyses, in which the duration of the included studies is variable – and not necessarily measured by commensurable metrics (e.g. some studies might use survival analyses, while others might use evaluations at a fixed point in time). 

Mean duration could be compared for some study pairs and did not seem to be consistently different in these; that said, we did not want to base any hard conclusions on those few studies, and chose not to include this analysis in the revised manuscript. That said, we did include a sentence in the discussion (page 23) to acknowledge the possibility that variation in these or other methodological issues could influence results, as suggested by the reviewer.

3. On page 25, the present analysis is judiciously compared with Ioannidis’ study from 2005. It would be helpful to know the median (and IQR) sample size of the index trials used in Ioannidis’ and those used here, because smaller trials are clearly less reliable than larger ones. It could well be that major journals have become more reluctant to publish smaller trials, even if they show surprisingly positive results. If so, the slightly higher replicability found in the present analysis could be interpreted as a possible improvement in publication policies (or citation patterns).

We have performed the analysis, which shows that the median sample size in Ioannidis’s sample is actually larger (median = 1500, IQR= 633 – 4382) than in ours (median = 332, IQR = 194 – 730). Thus, the difference (which may be partly due to the presence of cohort studies and to the lower prevalence of phase 1 studies (which were highly prevalent in our sample) does not seem to support the hypothesis brought up by the reviewer (i.e., that journals have become more reluctant to publish smaller trials). This is now mentioned in the discussion session (page 22).

Other changes:

- In the process of revising the manuscript, we realized that some highly cited studies had their sample sizes missing in our primary data file. This led to a minor revision in the third row of Table 6 (predictors of replication success) and Table S4, which does not qualitatively change the conclusions of our analysis (which remains underpowered to accurately detect predictors).

- Small clarifications or updated references were added to other parts of the manuscript (i.e., pages 23-25).

---

## [Decision Letter · Decision Letter 1]

2 Jul 2024

Estimating the replicability of highly cited clinical research (2004-2018)

PONE-D-23-26404R1

Dear Dr. da Costa,

We’re pleased to inform you that your manuscript has been judged scientifically suitable for publication and will be formally accepted for publication once it meets all outstanding technical requirements.

Kind regards,

Tim Mathes

Academic Editor

PLOS ONE

Additional Editor Comments (optional):

Reviewers' comments:

Reviewer's Responses to Questions

**Comments to the Author**

1. If the authors have adequately addressed your comments raised in a previous round of review and you feel that this manuscript is now acceptable for publication, you may indicate that here to bypass the “Comments to the Author” section, enter your conflict of interest statement in the “Confidential to Editor” section, and submit your "Accept" recommendation.

Reviewer #1: All comments have been addressed

Reviewer #2: All comments have been addressed

2. Is the manuscript technically sound, and do the data support the conclusions?

Reviewer #1: Yes

Reviewer #2: Yes

3. Has the statistical analysis been performed appropriately and rigorously? 

Reviewer #1: Yes

Reviewer #2: Yes

4. Have the authors made all data underlying the findings in their manuscript fully available?

Reviewer #1: Yes

Reviewer #2: Yes

5. Is the manuscript presented in an intelligible fashion and written in standard English?

Reviewer #1: Yes

Reviewer #2: Yes

6. Review Comments to the Author

Reviewer #1: The authors have addressed all my comments sufficiently. I am satisfied with the revision and recommend that the paper be accepted.

Reviewer #2: The reviewer thanks the authors for revising the manuscript in accordance to the reviewers' comments.

7. PLOS authors have the option to publish the peer review history of their article (what does this mean?). If published, this will include your full peer review and any attached files.

Reviewer #1: No

Reviewer #2: No

---

## [Editor Report · Acceptance letter]

16 Jul 2024

PONE-D-23-26404R1 

PLOS ONE

Dear Dr. da Costa, 

I'm pleased to inform you that your manuscript has been deemed suitable for publication in PLOS ONE. Congratulations! Your manuscript is now being handed over to our production team.

Kind regards, 

on behalf of

Dr. Tim Mathes 

Academic Editor

PLOS ONE